# Spray-Dried Powder Containing Chitinase and β-1,3-Glucanase with Insecticidal Activity against *Ceratitis capitata* (Diptera: Tephritidae)

**Bruno C. Aita [1], Silvana Schmaltz [1], Alex Fochi [1], Vinícius F. Bolson [1], Thiarles Brun [2], Lucas de Arruda Cavallin [3], Gabriel Camatti [3], Dori E. Nava [4], Jerson V. C. Guedes [3] , Raquel C. Kuhn [1], Giovani L. Zabot [5], Marcus Vinícius Tres [5],* and Marcio A. Mazutti [1]**

1 Department of Chemical Engineering, Federal University of Santa Maria, Santa Maria 97105-900, RS, Brazil; bcaita@hotmail.com (B.C.A.); silschmaltz@gmail.com (S.S.); fochi.alex@acad.ufsm.br (A.F.); bolson.vf@outlook.com (V.F.B.); raquelckuhn@yahoo.com.br (R.C.K.); marciomazutti@gmail.com (M.A.M.)

2 Department of Agricultural Engineering, Federal University of Santa Maria, Santa Maria 97105-900, RS, Brazil; brun.thiarles@gmail.com

3 Department of Plant Protection, Federal University of Santa Maria, Santa Maria 97105-900, RS, Brazil; lucas-cavallin@live.com (L.d.A.C.); camatti17@gmail.com (G.C.); jerson.guedes@gmail.com (J.V.C.G.)

4 Embrapa Clima Temperado, Entomology Laboratory, Pelotas 96010-971, RS, Brazil; dori.edson-nava@embrapa.br

5 Laboratory of Agroindustrial Processes Engineering (LAPE), Federal University of Santa Maria (UFSM), Cachoeira do Sul 96508-010, RS, Brazil; giovani.zabot@ufsm.br

* Correspondence: marcus.tres@ufsm.br; Tel.: +55-55-3220-9592

**Abstract:** This study focused on obtaining a spray-dried powder containing chitinase and β-1,3-glucanase as active ingredients for the control of agricultural pests. Different carriers were tested in the spray drying of these enzymes. The effectiveness of the application of the enzymes was evaluated against *Ceratitis capitata* (Diptera: Tephritidae). The combination of maltodextrin (2.5% *w/v*), gum Arabic (2.5% *w/v*), and soluble starch (5.0% *w/v*) as carriers showed the best result of residual activity of β-1,3-glucanase (88.36%) and chitinase (69.82%), with a powder recovery of 45.49%. The optimum conditions for the operational parameters of the spray drying process were: inlet air temperature of 120 °C, drying airflow rate of 1.1 m$^3$/min, feed flow rate of 5.8 mL/min, and nozzle air pressure of 0.4 MPa. The powder produced showed 65.6% efficiency for the control of the fly. These results demonstrated the possibility of using the spray drying process to obtain an enzymatic potential product for biological pest control.

**Keywords:** spray drying; cell wall degrading enzymes; biocontrol; microparticles

## 1. Introduction

The application of cell wall degrading enzymes for biological control of agricultural pests, mainly chitinases and β-1,3-glucanases, has increasingly received great attention [1–4]. However, the number of bioproducts containing enzymes as active ingredients for application in modern agriculture is still low, especially in Brazil, where no product is registered and authorized for commercialization [5]. The unavailability of these products in the market is caused by problems such as low enzymatic production and high purification costs during product development [6–8]. Another difficulty is maintaining enzyme stability and activity for a long time to allow storage and commercialization [9]. Enzymes are more susceptible to degradation and have a shorter shelf life when used in aqueous solutions. Therefore, it is preferable to store them as a dry powder [10]. Spray-drying is the most commonly used industrial technique to obtain dry powders since it is efficient, economical, and easily scalable [10,11].

A common concern during spray drying of enzyme-containing products is to avoid enzymatic thermal denaturation [12]. This can be achieved by optimizing the spray drying operating conditions and by the addition of stabilizing agents or carriers in the product [13]. In the spray drying of enzymes, carriers can act as a physical barrier that prevents protein denaturation against heat by different mechanisms such as hydrogen-bond formation with the protein molecules and kinetic stabilization due to immobilization of the enzyme in a glassy solid matrix [14,15]. The main carriers used in spray drying are polymers, polyols, and carbohydrates such as sucrose, trehalose, lactose, cyclodextrins, maltodextrins, gum arabic, and mannitol [10,14,16].

Several enzymes have been successfully spray-dried, such as lipase [15–17], β-galactosidase [18], β-fructofuranosidase [19], phytase [11], xylanase and amylase [10,12,20], savinase [21], phosphatase [22], peptidase [23], and inulinase [24]. However, after a scientific literature review in the main database, no reports could be found involving the spray drying of fungal chitinase and β-1,3-glucanase.

The Mediterranean fly *Ceratitis capitata* (Wiedemann) (Diptera: Tephritidae) is one of the main pests that affect fruit production worldwide due to its wide distribution, high polyphagia, and the fact that it causes high economic losses [25]. In Brazil, it is a major pest in the production of tropical fruits, especially in the São Francisco Valley, located between the states of Pernambuco and Bahia, where the largest fruit production hub in the country for export is located.

The main objective of the current study was to evaluate the spray drying of chitinase and β-1,3-glucanase produced by solid-state fermentation using the fungus *Metarhizium anisopliae*. Firstly, the effect of different carriers was evaluated on the physical-chemical properties of the powder obtained by spray drying of the enzymatic extracts. In the sequence, the operating conditions of the spray drying process were optimized to maximize the enzyme activity and the efficiency of the process. The powder obtained at optimized condition was used as a bioproduct for the control of *Ceratitis capitata* in vitro.

## 2. Materials and Methods

### 2.1. Enzyme Production and Extraction

The enzymes were produced by cultivating the fungus *Metarhizium anisopliae* (IBCB 348-Biological Institute of São Paulo, Brazil) by solid-state fermentation (SSF). Sugarcane bagasse was used as a substrate for SSF and the experimental conditions were detailed in the previous work reported by Aita et al. [26].

The fermented material was used for enzyme extraction using distilled water at a ratio of 1:10 (substrate:water). The flasks with fermented solid were stirred in an orbital shaker (New Brunswick, Model Innova 44, Brazil) at 150 rpm for 60 min. Then, the mixture was filtered with qualitative filter paper (Qualy™, J Problab, São José dos Pinhais, Brazil) and the liquid filtrate was centrifuged ($7000\times g$) at 4 °C for 10 min. The supernatant was filtered with microfiber filter paper (Chrom Filter™, B&C Biotech, Perugia, Italy) with a pore size of 0.22 μm. The filtered supernatant (enzyme extract) was used for the spray drying experiments. The enzymatic extract produced by SSF showed enzymatic activities of 33.12 U/g and 31.49 U/g of β-1,3-glucanase and chitinase, respectively, and a total solids content of 1% (*w/v*).

### 2.2. Spray Drying

The enzyme extract was dried in a spray dryer model LM MSDi 1.0 (Labmaq do Brasil Ltd.a., São Paulo, Brazil) with a drying capacity of 1.0 L/h and maximum air heating temperature of 180 °C. The dryer consisted of a stainless-steel drying chamber (0.67 m × 0.20 m) and cyclone (0.54 m × 0.095 m), and it was operated in concurrent flow with a two-fluid pneumatic nozzle (Labmaq do Brasil Ltd.a., São Paulo, Brazil) with 1.2 mm diameter. After spray drying, the dried powder was collected in 100 mL capped glass vials coupled at the bottom of the cyclone. The residual enzymatic activity (RA) of chitinase and β-1,3-glucanase, moisture content ($X_P$), recovery yield (Y), outlet air temperature

($T_{out}$), spray drying energetic efficiency (η), particle size distribution, and morphology were measured during the experiments. All experiments were analyzed in triplicate. The spray dryer was cleaned between each run.

In the first set of experiments, 10% (*w/v*) of the carrier was mixed with 50 mL of enzyme extract containing 1% (*w/v*) of total solids. The mixture was constantly mixed with a magnetic stirring bar and fed to the spray dryer nozzle by a peristaltic pump. The carriers evaluated were corn starch (Konkreta, Santa Catarina, Brazil), cassava starch (Konkreta, Santa Catarina, Brazil), soluble starch (Dinâmica, São Paulo, Brazil), maltodextrin DE 20 (Konkreta, Santa Catarina, Brazil), gum arabic (Dinâmica, São Paulo, Brazil), lactose (Dinâmica, São Paulo, Brazil), sucrose (Labsynth, São Paulo, Brazil), and mannitol (Sigma-Aldrich, São Paulo, Brazil). A control assay without the addition of a carrier was also performed. Different carrier proportions were also investigated. The operational parameters of the spray dryer were inlet air temperature of 120 °C, drying airflow rate of 1.1 $m^3$/min, feed flow rate of 5.8 mL/min, nozzle air pressure of 0.4 MPa, and nozzle airflow rate of 40 L/min.

In the second set of experiments, a 24 Central Composite Rotatable Design (CCRD) was used to optimize the spray drying operational parameters (feed flow rate—F, inlet air temperature—$T_{in}$, drying airflow rate—Q, and nozzle air pressure—P) on dryer performance ($T_{out}$; η; Y) and powder properties (RA of chitinase and β-1,3-glucanase; Xp). In all CCRD runs, the carriers maltodextrin, gum arabic, and soluble starch (2.5, 2.5, and 5.0% *w/v*, respectively) were mixed with 50 mL of enzyme extract and fed to the spray dryer nozzle using a peristaltic pump.

### 2.3. Measurement of Outlet Air Temperature, Energetic Efficiency, Powder Recovery, and Moisture Content

Outlet air temperature ($T_{out}$) was automatically measured by a probe connected to the spray dryer. The values were taken only after temperature stabilization. The spray dryer energetic efficiency (η) was calculated according to Equation (1) [27]:

$$\eta\ (\%) = \frac{T_{in} - T_{out}}{T_{in} - T_{room}} \times 100 \tag{1}$$

where $T_{in}$, $T_{out}$, and $T_{room}$ (°C) are inlet, outlet, and room temperatures, respectively.

Powder recovery yield (Y) was determined by Equation (2) [28]:

$$Y\ (\%) = \frac{W_{powder}}{W_{feed}} \times 100 \tag{2}$$

where $W_{powder}$ (g) is the weight of solids recovered after spray-drying and $W_{feed}$ (g) is the weight of solids in the feed solution. Powder moisture content ($X_p$) was determined after spray-drying using an infrared moisture analyzer (model IV 2500, Gehaka, São Paulo, Brazil).

### 2.4. Enzymatic Activities

The enzymatic activity of the powder recovered after spray drying was measured according to Suresh et al. [10]. For this, 0.1 g of spray-dried powder was added to 10 mL milli-Q water with 0.1% Tween-80 and mixed. The solution was used for chitinase and β-1,3-glucanase enzyme assays. Enzymatic activities of the enzyme extract before spray drying were determined. The RA of chitinase and β-1,3-glucanase was determined according to Equation (3) [16]:

$$RA\ (\%) = \frac{PA}{EA} \times 100 \tag{3}$$

where PA (U/mL) is the spray-dried powder enzymatic activity and EA (U/mL) is the enzymatic activity of the enzyme extract before spray drying.

Chitinase activity was determined according to Kim et al. [29] using colloidal chitin as substrate. β-1,3-glucanase activity was determined according to Jiang et al. [30] using laminarin (Sigma-Aldrich, São Paulo, Brazil) as substrate.

### 2.5. Scanning Electron Microscopy (SEM)

A scanning electron microscope (VEGA-3G, TESCAN, Brno, Czech Republic) was used to characterize the particle morphology in the spray drying experiments. Before the analysis, the samples were metallized with gold (spray metallization in an argon atmosphere, using an electric current of 20 mA for 90 s).

### 2.6. Particle Size Distribution

The particle size distribution of the powder obtained by spray drying was determined by laser diffraction using a Laser Scattering Spectrometer Mastersizer 2000 (Malvern Instruments, Malvern, Worcestershire, UK), equipped with a dispersion unit and sample introduction model Hydro 2000S (Malvern Instruments). The rotation speed was 1700 rpm for introducing the sample and 500 rpm for the measurements. Vaseline was used as a dispersant medium. The measurements were performed in quintuplicate.

### 2.7. Insecticidal Activity of the Powder Obtained by Spray Drying

Pupae of Mediterranean fruit fly (*C. capitata*) were reared according to Ricalde et al. [31]. The insects used were obtained from the entomology laboratory at Embrapa Temperate Agriculture (Pelotas, Rio Grande do Sul, Brazil). The powder obtained with the spray drying optimized conditions was evaluated for its insecticidal activity against *C. capitata*. The assays were performed according to susceptibility method number 30 of the Insecticide Resistance Action Committee [32]. Three concentrations of the powder were evaluated separately: 50 g/L (treatment 1), 100 g/L (treatment 2), and 200 g/L (treatment 3). The solutions were prepared by diluting the powder in distilled water and mixing for 30 min. The control treatment was prepared with the application of distilled water instead of the solution with the spray-dried powder. The assays were performed in quadruplicate.

Aerated acrylic boxes (11 × 11 × 3 cm) were used in the assays, in which 1 mL of the powder solution or distilled water (control) was uniformly deposited. Then, nine pupae of *C. capitata* were deposited in each container. Control efficiency assessments were carried out 24 h after pupae emergence to the adult stage. The containers containing the flies were kept throughout the evaluation period at 28 °C and the control efficiency was determined according to Equation (4) [33]:

$$\text{Control efficiency (\%)} = \frac{A - B}{A} \times 100 \tag{4}$$

where A is the number of living insects in the control sample and B is the number of living insects in each treatment with the spray-dried powder.

### 2.8. Statistical Analysis

Statistical analysis of the experimental data was performed using the software Statistica® 7.0 (Statsoft Inc., Tulsa, OK, USA). Analysis of variance (ANOVA) with a significance level of 90% (*p*-value < 0.10) was used to evaluate the significance between different experimental conditions.

## 3. Results and Discussion

### 3.1. Effect of Different Carriers on the Spray Drying of Enzyme Extract

The RA of β-1,3-glucanase ranged from 43.61–91.10% (Table 1). The highest RA of β-1,3-glucanase was obtained with gum arabic (91.10%) and maltodextrin DE20 (87.90%). Chitinase was more susceptible to loss of enzymatic activity during the spray drying process when compared to β-1,3-glucanase and the RA of chitinase ranged from 0 to 99.70%. These results are in agreement with other studies where the drying of enzymatic solutions by

spray drying resulted in residual enzyme activity values from 87.2% to 100% for lipase with the carriers lactose, maltodextrin DE20, and gum arabic [17], 90 to 95% for xylanase using malt extract, lactose and maltodextrin [12], and 100% for alkaline proteases with the use of maltodextrin [34].

**Table 1.** Effect of different carriers on the spray drying of enzyme extract.

| Carriers * | $T_{out}$ (°C) | $X_p$ (%) | RA (%) | | Y (%) | η (%) |
| | | | β-1,3-Glucanase | Chitinase | | |
| --- | --- | --- | --- | --- | --- | --- |
| Corn starch | 77.0 ± 2.0 [cde] | 4.05 ± 0.07 [a] | 43.84 ± 2.99 [d] | 0.01 ± 0.01 [d] | 40.22 ± 1.88 [b] | 45.03 ± 1.81 [cde] |
| Cassava starch | 79.0 ± 0.1 [e] | 4.35 ± 0.04 [ab] | 57.53 ± 9.42 [cd] | 9.31 ± 2.27 [cd] | 29.67 ± 0.39 [c] | 42.93 ± 0.01 [e] |
| Soluble starch | 78.0 ± 0.1 [de] | 3.75 ± 0.07 [a] | 43.61 ± 2.20 [d] | 0.01 ± 0.01 [d] | 59.13 ± 4.28 [a] | 43.98 ± 0.01 [de] |
| Maltodextrin | 76.0 ± 1.7 [bcd] | 7.90 ± 0.92 [b] | 87.90 ± 4.67 [a] | 99.70 ± 12.8 [a] | 11.27 ± 0.21 [d] | 46.07 ± 1.81 [bcd] |
| Gum arabic | 73.0 ± 0.6 [a] | 5.63 ± 0.23 [ab] | 91.10 ± 10.76 [a] | 60.66 ± 1.04 [b] | 13.44 ± 1.54 [d] | 48.87 ± 0.60 [a] |
| Lactose | 74.0 ± 1.0 [ab] | 5.18 ± 0.67 [ab] | 69.95 ± 3.84 [bc] | 56.06 ± 4.07 [b] | 13.48 ± 0.95 [d] | 48.17 ± 1.05 [ab] |
| Sucrose | 75.0 ± 0.6 [abc] | 12.55 ± 0.71 [c] | 57.99 ± 7.51 [cd] | 16.10 ± 1.81 [c] | 9.35 ± 1.22d [e] | 46.77 ± 0.60 [abc] |
| Mannitol | 76.0 ± 1.2 [bcd] | 17.36 ± 5.61 [d] | 50.00 ± 4.79 [d] | 0.01 ± 0.01 [d] | 5.95 ± 0.76 [e] | 45.72 ± 1.21 [bcd] |
| Without carriers | 78.0 ± 0.1 [de] | 45.1 ± 0.10 [e] | 84.11 ± 1.90 [ab] | 0.01 ± 0.01 [d] | 1.60 ± 0.03 [e] | 43.98 ± 0.01 [de] |

$T_{out}$: outlet air temperature; $X_p$: powder moisture content; RA: powder residual enzymatic activity; Y: powder recovery yield; η: spray dryer energetic efficiency. Means in the same column with different lowercase superscript letters ([a, b, c, d, e]) are significantly different from each other by Tukey's test (*p*-value < 0.10). * All carriers were used at a concentration of 10% (*w/v*).

The mechanisms of enzymatic inactivation in spray drying processes have not been yet fully studied, but it is known that inactivation involves conformational changes in the structure of protein molecules, such as unfolding and aggregation [14,15]. A more rigid native conformation of the protein causes difficulty to unfold its structure and, consequently, more difficulty to destroy its catalytic center [35]. Probably, β-1,3-glucanase produced by *M. anisopliae* IBCB 348 has a more rigid structure, and it was more able to resist the high drying temperatures and shear stresses of spray drying than chitinase. The different affinity of each enzyme with the molecules of the carrier is another factor that may explain the differences in enzymatic inactivation between β-1,3-glucanase and chitinase, but this needs to be confirmed by further studies.

The nature of the carbohydrate used as carriers affects enzyme preservation during spray drying through the formation of hydrogen bonds between the dried proteins and the carbohydrates and kinetic stabilization due to the immobilization of the enzyme in a rigid and glassy matrix, which is essential to avoid alterations in the protein structure [14,15,19,36]. In spray drying processes, the use of carriers with high glass transition temperatures is generally required [37]. The highest RA values of β-1,3-glucanase and chitinase obtained with maltodextrin DE20 and gum arabic can be attributed, mainly, to the high glass transition temperature of these two carriers of 141 °C and 126 °C, respectively [38,39]. The ability of hydroxyl groups present in the composition of these carriers to form hydrogen bonds with the enzyme extract proteins, replacing the water molecules lost during drying and avoiding its thermal denaturation, is another important factor that could explain the higher retention of enzymatic activity with gum arabic and maltodextrin [40].

The results also indicated that the RA of chitinase was highly dependent on the type of carrier. The use of maltodextrin as a carrier was able to maintain practically all enzymatic activity (99.70%), followed by gum arabic (60.66%) and lactose (56.06%). All chitinolytic activity was lost with corn starch, soluble starch, and mannitol, which also showed the lowest RA of β-1,3-glucanase. Soluble starch and corn starch have high glass transition temperatures, of approximately 245 °C [41], which are much higher than the spray dryer inlet air temperature (120 °C). The high temperatures hamper agglomeration and excess viscosity of the final product, which is necessary to the process. However, the loss of residual enzyme activity may have occurred due to the low availability of hydroxyl groups

in the starch molecules. Taylor and Zografi [42] reported that the formation of hydrogen bonds is inversely correlated to the glass transition temperature of the carbohydrate, with a decreased tendency to form bonds as the glass transition temperature increases. According to Millqvist-Fureby et al. [36], proteins that do not bind with the carrier molecules accumulate on the surface of the droplets sprayed in the spray dryer, causing high protein exposure to high temperatures and loss of enzymatic activity.

The morphological analysis of the particles by Scanning Electron Microscopy (SEM) (Figures 1 and 2) was used to investigate the relationship between the carrier and the preservation of enzymatic activity. The interaction of the enzymatic extract with all starches resulted in the formation of microparticles with a relatively smooth surface, with some irregularities. Soluble starch (Figure 1C1,C2) enabled microparticles to have a more defined round shape than with corn starch (Figure 1A1,A2) and cassava starch (Figure 1B1,B2). As previously mentioned, a hypothesis for the low RA of β-1,3-glucanase and chitinase obtained with starches, even with the formation of well-defined granules, was that the proteins in the enzyme extract remained on the granule surface, which resulted in their denaturation. Some authors reported a decrease in enzyme activity and identified that the surface of the powders produced by spray drying of proteins and carbohydrates mixtures have a high surface accumulation of proteins due to the adsorption of proteins in the air:water interface in the spray droplets before solvent evaporation [36,43].

The microparticles formed with maltodextrin (Figure 2D1,D2) and gum arabic (Figure 2E1,E2) were smaller than with starches. The microparticles formed with maltodextrin had a rounded shape and a smooth surface, whereas those formed with gum arabic had an indefinite shape and the surface was wrinkled. According to Cai and Corke [44], granules with smooth walls or without cracks have higher protection of the core to heat and oxidation. Therefore, the proteins most likely remained in the innermost part of the maltodextrin and gum arabic granules, maintaining the enzymatic activity.

A higher RA of β-1,3-glucanase (84.11%) was observed in the dried enzymatic extract without the addition of carriers, which indicates the thermostability of this enzyme during the drying process. However, the enzyme extract without the addition of carriers completely lost the RA of chitinase, probably because of the higher sensitivity of this enzyme to the spray drying conditions. Despite having formed a final product with high RA of β-1,3-glucanase, spray drying of enzymatic extract without the addition of carriers formed an extremely low amount of powder, with a powder recovery yield of only 1.60% and a very high moisture content (45.1%), which makes it impossible to obtain a suitable final product.

The powder recovery yield (Y) ranged widely according to the carrier type (Table 1). The best results of Y were obtained with soluble starch (59.13%), followed by corn starch (40.22%) and cassava starch (29.67%). Yields in this range (20 to 60%) are considered common for laboratory-scale spray dryers [45–47]. Yields below 15% were obtained with the other carriers, which may have occurred due to the inefficient dust collection system of the spray dryer, which leads to a loss of fine particles with the exhaust air. The particle size distribution of the powder with each carrier was performed to demonstrate this hypothesis (Table 2). The particle size distribution depended heavily on the carrier, with an average diameter (D0.5) that ranged from 8.39 to 95.97 μm. The highest D0.5 was obtained with the use of starch, which reduced powder losses with the exhaust air in the spray dryer cyclone and resulted in the highest powder recovery yield. The best results in the retention of enzymatic activity were obtained with maltodextrin and gum arabic. However, the deposition and adherence of the powder in the drying chamber and the spray dryer cyclone resulted in low Y (11.27 and 13.44%, respectively). This could be explained because the final product obtained with these carriers had higher moisture content than starches (Table 1). The moisture content of the final product ($X_p$) reduces its glass transition temperature and can lead to the material sticking to the equipment surfaces [11].

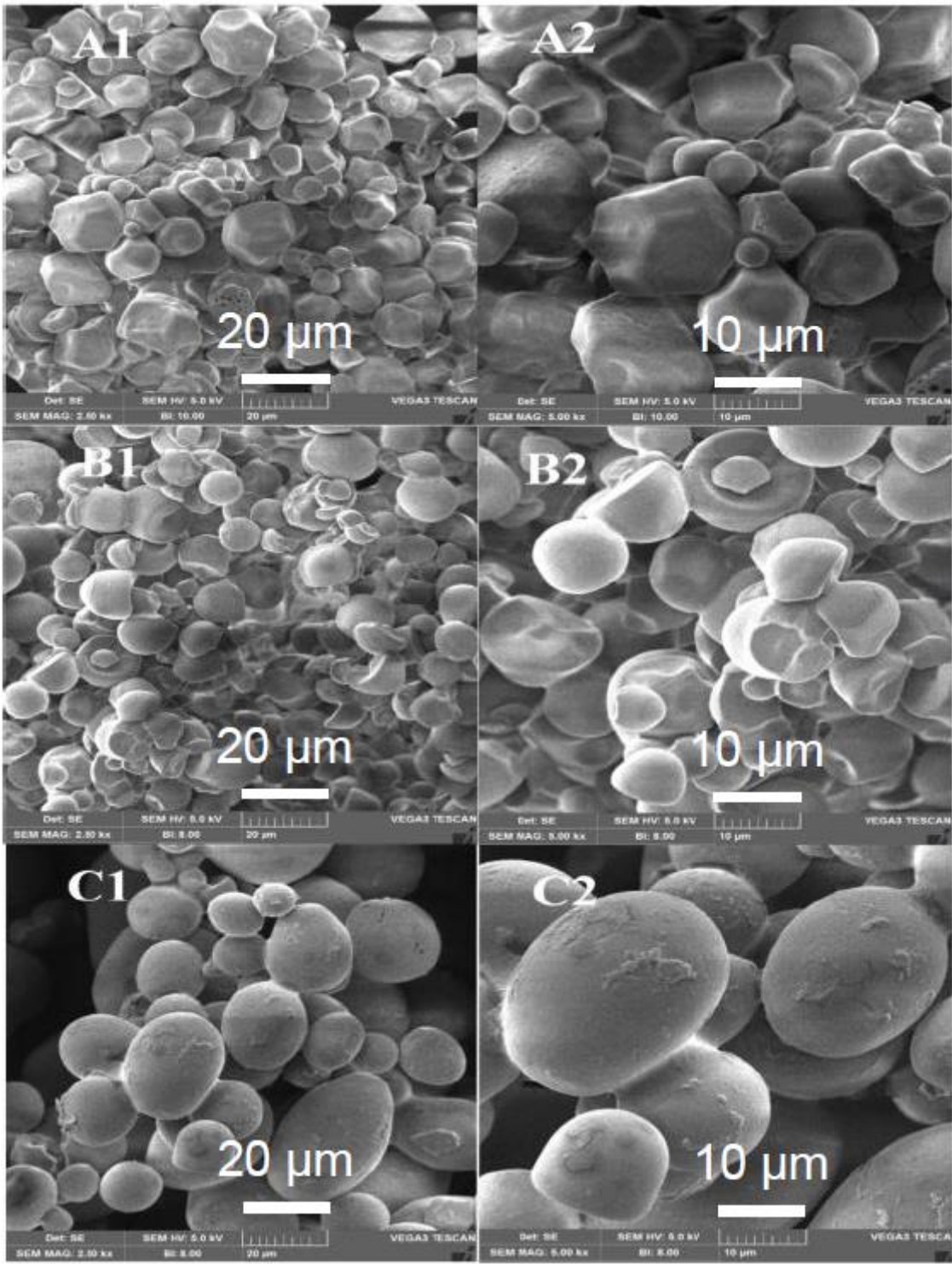

**Figure 1.** Scanning electron photomicrographs of the powders formed in the spray drying of the enzymatic extract with the carriers: corn starch (**A1,A2**), cassava starch (**B1,B2**), and soluble starch (**C1,C2**). (**A1,B1,C1**): 2.5 kX magnification; (**A2,B2,C2**): 5.0 kX magnification.

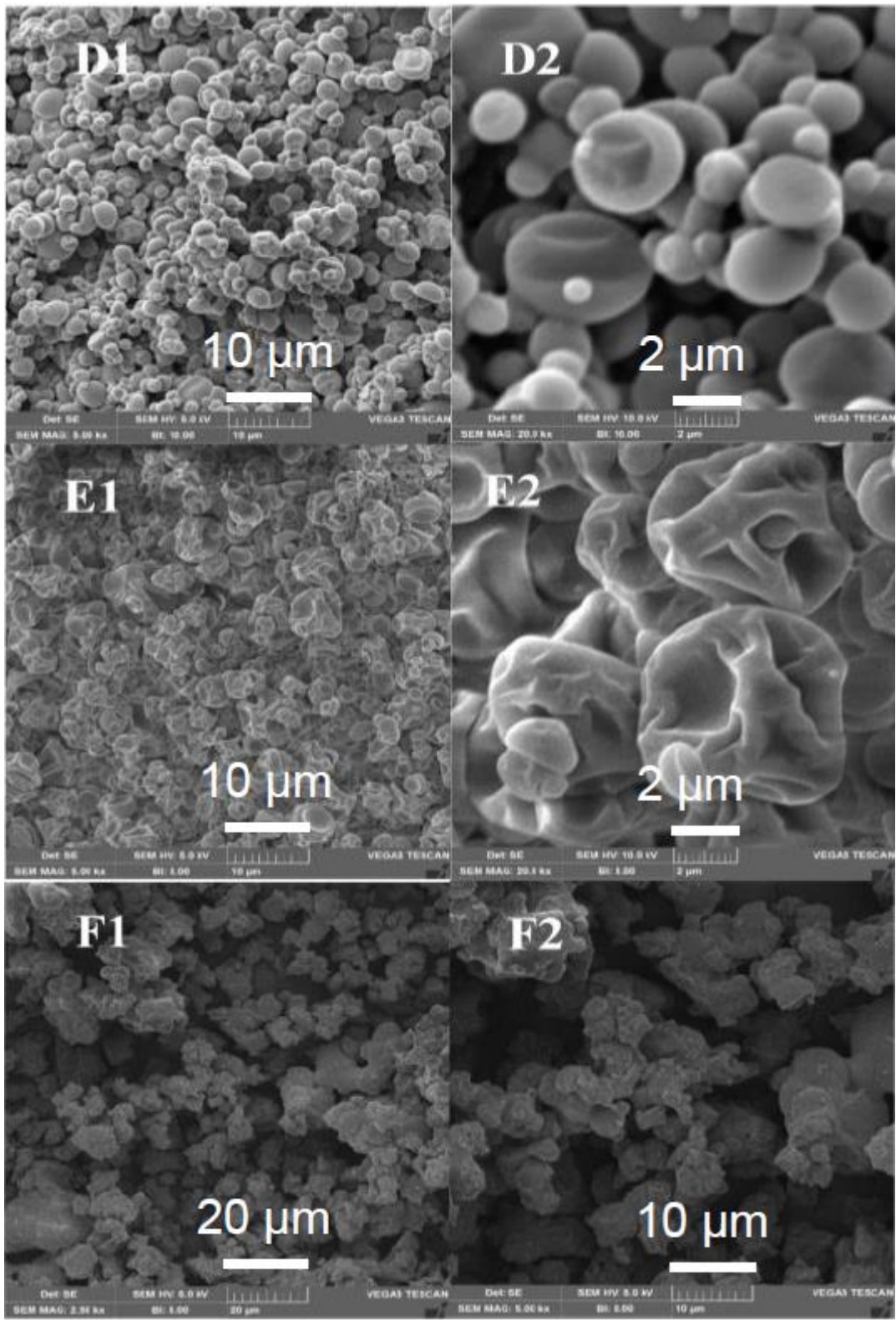

**Figure 2.** Scanning electron photomicrographs of the powders formed in the spray drying of the enzymatic extract with the carriers: maltodextrin (**D1**,**D2**), gum arabic (**E1**,**E2**) and mannitol (**F1**,**F2**). (**D1**,**E1**,**F2**): 5.0 kX magnification; (**F1**): 2.5 kX magnification; (**D2**,**E2**): 20.0 kX magnification.

**Table 2.** Particle size distribution of the powders produced with each carrier by spray drying.

| Carriers | $D_{0,1}$ (µm) | $D_{0,5}$ (µm) | $D_{0,9}$ (µm) |
|---|---|---|---|
| Corn starch | 10.46 | 39.93 | 267.05 |
| Cassava starch | 6.94 | 26.10 | 143.92 |
| Soluble starch | 27.05 | 95.97 | 251.87 |
| Maltodextrin | 4.45 | 15.82 | 218.45 |
| Gum arabic | 3.02 | 8.39 | 17.33 |
| Lactose | ND | ND | ND |
| Sucrose | ND | ND | ND |
| Mannitol | 15.16 | 50.91 | 113.50 |

D: Mean diameters of the distribution curve accumulated in 10% ($D_{0,1}$), 50% ($D_{0,5}$), and 90% ($D_{0,9}$) of the sample total volume. ND: Not determined due to the sample high hygroscopicity.

The Xp using different carriers ranged from 3.75 to 17.36% and, except for sucrose and mannitol, it was below 8%. In drying processes such as spray drying, Xp is an indicator of quality, as the presence of water influences the integrity of the solid matrix and the protein–carrier interactions [16]. According to Bone [48], water can cause oscillatory and rotational movement of amino acids groups in proteins and also segmental and internal fluctuations that increase protein dynamic mobility, decreasing its conformational stability. Therefore, low Xp was considered adequate for the stability of the final product. Although Chang et al. [49] indicate Xp of 2% to 3% as ideal for the storage of dry powders containing proteins, these values must be evaluated for each type of protein separately.

During spray drying, the temperature reached by the final product was close to the outlet air temperature ($T_{out}$). Therefore, the determination of this temperature is important to evaluate the degradation of heat-sensitive enzymes [50]. Tout ranged from 73 to 79 °C (Table 1), indicating the reduced influence of carriers on this parameter. The spray dryer energetic efficiency (η) is often used to evaluate the performance and energy consumption of spray drying, as it indicates the efficiency in heat transfer between the drying air and the atomized fluid [51]. The η was practically independent of the type of carrier and ranged from 42.93 to 48.87% (Table 1). These results were expected because the drying operational conditions were the same between the assays carried out with each carrier and η depends basically on the temperature of the drying air (inlet and outlet) and the ambient temperature. The values of η obtained in this study were in agreement with other research, such as that by Santana et al. [51], in the spray drying of Pequi pulp (29.9 to 44.8%) and Cortés-Rojas et al. [27] in the spray drying of *Bidens pilosa* L. (31.0 to 51.8%).

### 3.2. Different Carrier Combinations in the Spray Drying of Enzymatic Extract

The selection of the carrier in spray drying is necessary to maximize enzymatic production, which depends on a high residual enzymatic activity and yield. Some carriers that provided powder with the highest residual enzymatic activity were not the same as the ones with the higher yield. Therefore, different carrier combinations were evaluated. The carriers that provided the highest RA (maltodextrin, gum arabic, and lactose) were mixed with the carrier that showed the highest Y (soluble starch) in different proportions (Table 3).

The carrier combinations made it possible to obtain a high Y and to guarantee the residual enzymatic activity of β-1,3-glucanase and chitinase. The carrier combination that showed the best result was maltodextrin (2.5% *w/v*), gum arabic (2.5% *w/v*), and soluble starch (5.0% *w/v*) in run 6, with an RA of β-1,3-glucanase and chitinase of 88.36 and 69.82%, respectively, and a Y of 45.49%. These results indicate the synergistic effect of the carriers. The soluble starch reduced the adherence of the powder on the equipment due to its higher glass transition temperature and the gum arabic and maltodextrin allowed a higher retention of proteins on its structure.

**Table 3.** Effect of different carrier combinations and concentrations in the spray drying of enzyme extract.

| Run | Carriers | Conc. (%, *w/v*) | $T_{out}$ (°C) | $X_p$ (%) | RA (%) | | Y (%) | H (%) |
|---|---|---|---|---|---|---|---|---|
| | | | | | β-1,3-Glucanase | Chitinase | | |
| 1 | Maltodextrin<br>Soluble starch | 5.0<br>5.0 | 74.0 ± 0.1 [b] | 3.75 ± 0.07 [ab] | 59.93 ± 0.48 [c] | 20.27 ± 3.19 [e] | 46.18 ± 0.78 [a] | 46.00 ± 0.01 [a] |
| 2 | Gum arabic<br>Soluble starch | 5.0<br>5.0 | 74.0 ± 0.1 [b] | 4.55 ± 0.64 [b] | 77.05 ± 1.45 [b] | 33.78 ± 0.64 [d] | 45.37 ± 0.15 [a] | 46.00 ± 0.01 [a] |
| 3 | Gum arabic<br>Soluble starch | 2.5<br>7.5 | 75.0 ± 0.1 [a] | 2.65 ± 0.35 [a] | 79.79 ± 3.39 [ab] | 56.76 ± 1.27 [b] | 36.24 ± 4.36 [b] | 45.00 ± 0.01 [b] |
| 4 | Lactose<br>Soluble starch | 5.0<br>5.0 | 74.0 ± 0.1 [b] | 3.75 ± 0.49 [ab] | 64.38 ± 1.94 [c] | 21.27 ± 0.67 [e] | 48.15 ± 0.43 [a] | 46.00 ± 0.01 [a] |
| 6 | Maltodextrin<br>Gum arabic<br>Soluble starch | 2.5<br>2.5<br>5.0 | 75.0 ± 0.1 [a] | 3.75 ± 0.21 [ab] | 88.36 ± 4.84 [a] | 69.82 ± 3.19 [a] | 45.49 ± 0.29[a] | 45.00 ± 0.01 [b] |
| 7 | Maltodextrin<br>Gum arabic<br>Lactose<br>Soluble starch | 2.5<br>2.5<br>2.5<br>2.5 | 74.0 ± 0.1 [b] | 2.95 ± 0.21 [a] | 33.90 ± 0.48 [d] | 41.89 ± 0.66 [c] | 50.19 ± 3.16 [a] | 46.00 ± 0.01 [a] |

$T_{out}$: outlet air temperature; $X_p$: powder moisture content; RA: powder residual enzymatic activity; Y: powder recovery yield; η: spray dryer energetic efficiency. Means in the same column with different lowercase superscript letters ([a, b, c, d, e]) are significantly different from each other by Tukey's test (*p*-value < 0.10).

The morphological analysis of the powder obtained with the carrier combination (Figure 3) indicates that the particles of each carrier in the mixture were similar to the particles formed with each carrier individually (Figures 1 and 2). In addition, Figure 3M2 clearly shows the adhesion of the smaller particles of maltodextrin and gum arabic on the surface of the starch granules (larger particles). The powder obtained with the carrier combination had a D0.5 of 39.93 μm, which is higher than those ones obtained with maltodextrin (15.82 μm) and gum arabic (8.39 μm), reducing the loss of fine particulate material with the exhaust air. Another positive effect of the carrier combination was on the $X_p$, which was below 5%.

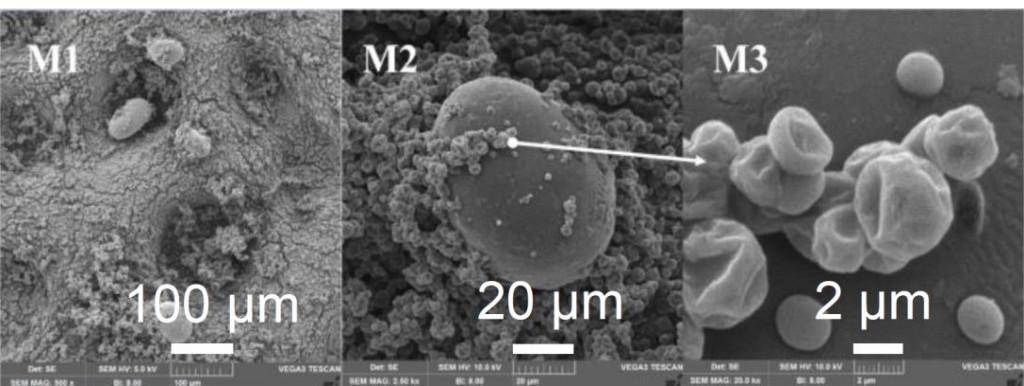

**Figure 3.** Scanning electron photomicrographs of the powders formed in the spray drying of the mixture of maltodextrin (2.5% *w/v*); gum arabic (2.5% *w/v*); soluble starch (5.0% *w/v*) with enzymatic extract (**M1–M3**). The white arrows indicate the point where the magnification was performed (20 kx).

### 3.3. Spray Drying Optimization

All optimization runs (CCRD 24) were carried out with the mixture of enzyme extract with maltodextrin (2.5% *w/v*), gum arabic (2.5% *w/v*), and soluble starch (5.0% *w/v*). This was the carrier combination that provided the best results in previous experiments. The results in the CCRD indicated that the RA was highly dependent on the operational conditions of the spray drying process (Table 4). The RA for β-1,3-glucanase ranged from 26.71 (run 18) to 102.05% (run 19) and for chitinase ranged from 0 (run 15) to 55.86% (run 27).

The Y ranged from 26.9 (run 23) to 46.68% (run 8). The recovered powder presented X*p* between 1.8 (run 21) and 6.6% (run 3), while Tout ranged from 62 (run 4) to 91 °C (run 13).

**Table 4.** Experimental design and results obtained in the spray drying optimization.

| Run | $T_{in}$ (°C) | Q (m³/min) | F (mL/min) | P (MPa) | RA (%) | | Y (%) | $X_p$ (%) | $T_{out}$ (°C) |
| | | | | | β-1,3-Glucanase | Chitinase | | | |
|---|---|---|---|---|---|---|---|---|---|
| 1 | 110 (−1) | 0.9 (−1) | 5.0 (−1) | 0.3 (−1) | 83.56 | 14.41 | 31.18 | 4.4 | 69.0 |
| 2 | 110 (−1) | 0.9 (−1) | 5.0 (−1) | 0.5 (1) | 91.78 | 27.93 | 37.32 | 4.8 | 66.0 |
| 3 | 110 (−1) | 0.9 (−1) | 6.6 (1) | 0.3 (−1) | 85.62 | 23.42 | 27.90 | 6.6 | 64.0 |
| 4 | 110 (−1) | 0.9 (−1) | 6.6 (1) | 0.5 (1) | 82.88 | 37.84 | 31.83 | 5.5 | 62.0 |
| 5 | 110 (−1) | 1.3 (1) | 5.0 (−1) | 0.3 (−1) | 85.62 | 39.64 | 28.35 | 3.6 | 74.0 |
| 6 | 110 (−1) | 1.3 (1) | 5.0 (−1) | 0.5 (1) | 82.19 | 39.64 | 42.71 | 3.0 | 73.0 |
| 7 | 110 (−1) | 1.3 (1) | 6.6 (1) | 0.3 (−1) | 77.40 | 27.93 | 32.18 | 4.2 | 75.0 |
| 8 | 110 (−1) | 1.3 (1) | 6.6 (1) | 0.5 (1) | 73.97 | 23.42 | 46.68 | 3.4 | 74.0 |
| 9 | 130 (1) | 0.9 (−1) | 5.0 (−1) | 0.3 (−1) | 69.86 | 35.14 | 34.34 | 3.8 | 79.0 |
| 10 | 130 (1) | 0.9 (−1) | 5.0 (−1) | 0.5 (1) | 93.15 | 50.45 | 37.36 | 3.8 | 77.0 |
| 11 | 130 (1) | 0.9 (−1) | 6.6 (1) | 0.3 (−1) | 41.78 | 30.63 | 28.60 | 3.6 | 75.0 |
| 12 | 130 (1) | 0.9 (−1) | 6.6 (1) | 0.5 (1) | 56.16 | 32.43 | 33.84 | 4.7 | 74.0 |
| 13 | 130 (1) | 1.3 (1) | 5.0 (−1) | 0.3 (−1) | 63.01 | 16.22 | 37.30 | 3.1 | 91.0 |
| 14 | 130 (1) | 1.3 (1) | 5.0 (−1) | 0.5 (1) | 75.34 | 16.22 | 45.80 | 2.2 | 87.0 |
| 15 | 130 (1) | 1.3 (1) | 6.6 (1) | 0.3 (−1) | 67.12 | 0.00 | 37.53 | 2.3 | 87.0 |
| 16 | 130 (1) | 1.3 (1) | 6.6 (1) | 0.5 (1) | 62.33 | 0.90 | 42.20 | 3.2 | 84.0 |
| 17 | 100 (−2) | 1.1 (0) | 5.8 (0) | 0.4 (0) | 72.60 | 21.62 | 35.53 | 4.7 | 64.0 |
| 18 | 140 (2) | 1.1 (0) | 5.8 (0) | 0.4 (0) | 26.71 | 8.11 | 39.55 | 2.4 | 86.0 |
| 19 | 120 (0) | 0.7 (−2) | 5.8 (0) | 0.4 (0) | 102.05 | 39.64 | 35.25 | 5.1 | 65.0 |
| 20 | 120 (0) | 1.5 (2) | 5.8 (0) | 0.4 (0) | 86.99 | 18.02 | 43.33 | 2.8 | 80.0 |
| 21 | 120 (0) | 1.1 (0) | 4.2 (−2) | 0.4 (0) | 87.67 | 48.65 | 42.23 | 1.8 | 78.0 |
| 22 | 120 (0) | 1.1 (0) | 7.4 (2) | 0.4 (0) | 86.30 | 46.85 | 35.69 | 4.4 | 73.0 |
| 23 | 120 (0) | 1.1 (0) | 5.8 (0) | 0.2 (−2) | 92.47 | 34.23 | 26.90 | 6.3 | 78.0 |
| 24 | 120 (0) | 1.1 (0) | 5.8 (0) | 0.6 (2) | 80.82 | 39.64 | 40.92 | 4.4 | 72.0 |
| 25 | 120 (0) | 1.1 (0) | 5.8 (0) | 0.4 (0) | 93.84 | 53.15 | 40.23 | 4.1 | 74.0 |
| 26 | 120 (0) | 1.1 (0) | 5.8 (0) | 0.4 (0) | 93.84 | 54.95 | 37.54 | 4.2 | 74.0 |
| 27 | 120 (0) | 1.1 (0) | 5.8 (0) | 0.4 (0) | 94.52 | 55.86 | 38.75 | 3.9 | 75.0 |

$T_{in}$: inlet air temperature; Q: drying airflow rate; F: feed flow rate; P: nozzle air pressure; RA: powder residual enzymatic activity; Y: powder recovery yield; $X_p$: powder moisture content; $T_{out}$: outlet air temperature.

The effects of the independent variables ($T_{in}$, F, Q, and P) were estimated from the experimental results obtained in the CCRD. Codified models were elaborated (Table 5) considering only the independent variables that presented statistically significant effects ($p < 0.10$). The models were validated by the ANOVA, Fisher's test (F-test), and coefficient of determination ($R^2$).

The residual enzymatic activity of β-1,3-glucanase was reduced with the increase in Tin and the lowest value (26.71%) was obtained in the run with the highest Tin (140 °C). Since high temperatures can cause enzyme denaturation, this was already expected. Similar results were found in the spray drying of bromelain with a Tin ranging from 100 to 120 °C [13] and in the spray drying of amylase from 130 to 230 °C [20]. The increase in the feed flow rate had a negative effect on the residual enzymatic activity of β-1,3-glucanase. This occurred because in lower feed flow rates the material is dried faster since there is less liquid to be dried. The formation of a carrier crust with a protective effect on proteins occurs more quickly with a higher rate of water evaporation [27].

**Table 5.** Adjusted models of the Central Composite Rotatable Design (CCRD) dependent variables.

| Model Equations | Coefficient of Determination and Fisher's Test Values |
|---|---|
| $RA_{\beta\text{-1,3-glucanase}} = 87.59 - 9.42T_{in} - 10.73(T_{in})^2 - 4.34F$ | $R^2 = 0.75$ $F_{calc} = 22.73$ $F_{tab} = 2.34$ |
| $RA_{chitinase} = 54.65 - 3.36T_{in} - 10.94(T_{in})^2 - 5.55Q - 7.45(Q)^2 - 2.88F - 2.91(F)^2 + 2.18P - 5.42(P)^2 - 8.90(T_{in}Q) - 2.81(T_{in}F) - 3.49(QF) - 3.04(QP)$ | $R^2 = 0.94$ $F_{calc} = 17.20$ $F_{tab} = 2.05$ |
| $Y = 37.83 + 1.12T_{in} + 2.80Q - 1.13F + 3.68P - 1.29(P)^2 - 1.09(T_{in}P) + 1.42(QF) + 1.48(QP)$ | $R^2 = 0.91$ $F_{calc} = 23.17$ $F_{tab} = 2.04$ |
| $X_p = 3.84 - 0.56T_{in} - 0.70Q + 0.42F - 0.23(F)^2 + 0.35(P)^2$ | $R^2 = 0.80$ $F_{calc} = 16.80$ $F_{tab} = 2.14$ |
| $T_{out} = 75.16 + 5.88T_{in} + 4.56Q - 1.28F - 1.21P + 0.71(QF)$ | $R^2 = 0.96$ $F_{calc} = 111.70$ $F_{tab} = 2.14$ |

$RA_{\beta\text{-1,3-glucanase}}$: β-1,3-Glucanase residual enzymatic activity; $RA_{chitinase}$: chitinase residual enzymatic activity; Y: powder recovery yield; $X_p$: powder moisture content; $T_{out}$: outlet air temperature; F: feed flow rate coded value; $T_{in}$: inlet air temperature coded value; Q: drying air flow rate coded value; P: nozzle air pressure coded value. $R^2$: coefficient of determination; $F_{calc}$: F calculated value of the Fisher's test; $F_{tab}$: F table value of the Fisher's test.

An increase in the Tin and in the Q reduced the residual enzymatic activity of chitinase. In runs 15 and 16 (Tin of 130 °C and Q of 1.3 m$^3$/min), the enzymatic activity of chitinase was completely inhibited. The flow rate of the drying air determines the rate of heat transfer and the residence time of the droplets/powder particles in spray drying processes [52]. The residence time (ratio between the volume of the drying chamber and the Q) was 0.83 and 1.8 s with the maximum and lowest Q, respectively. Although it reduces the time of product exposure to high temperatures, the use of high Q can increase the heat transfer and cause the thermal denaturation of chitinase.

Higher Y values were obtained by increasing the $T_{in}$ and decreasing the feed flow rate. Maury et al. [52] and Broadhead et al. [45] evaluated the influence of operational parameters on the spray drying of trehalose and β-galactosidase, respectively. The authors explained that it was possible to obtain a higher Y using higher drying temperatures and reducing the flow of liquid that was fed in the process. According to the authors, this behavior was a consequence of the reduction in the powder moisture content because the reduction in feed flow rate indicates a lesser amount of liquid to be evaporated in s specific time, while higher temperatures facilitate water evaporation. In spray dryers with two fluid atomizers, the droplets are sprayed towards the vertical wall of the drying chamber. If the drop/particle was not completely dried before reached the wall, the particles could stick to the wall, resulting in the formation of a wet deposit and reducing final product recovery [52]. This phenomenon was more evident in the results showed by Table 4, in which the run with the lowest yield were those with the highest $X_p$ (runs 3 and 23).

Higher yields were obtained when increasing the drying air flow rate because a higher drying airflow increases both the rate of particle separation in the cyclone and the drag forces in the drying chamber, improving powder recovery [53]. The increase in P resulted in higher yields since higher P leads to the formation of smaller droplets, which contributes to the drying process and prevents the yield reduction by wet particles adherence on the dryer walls [54].

The runs that resulted in the formation of powder with a low $X_p$ were the same as high $T_{out}$. The models from Table 5 show that an increase in $T_{in}$ and Q results in a decrease in $X_p$ and an increase in $T_{out}$. This was expected, considering that a more efficient drying can be achieved by increasing the thermal energy supplied to the system. Similarly, higher values of F reduced $T_{out}$ and increased $X_p$ due to the higher volume of liquid to be evaporated. Powder with an $X_p$ below 2% was obtained with maximum $T_{in}$ (140 °C) and Q (1.5 m$^3$/min), a moisture content considered excellent for spray drying. However,

enzymatic inactivation was observed using those values of $T_{in}$ and Q, and this must be considered in the optimized condition.

Simultaneous Optimization of Spray Drying Using the Desirability Function

The spray drying optimization depends on the simultaneous evaluation of five response variables (RA of β-1,3-glucanase, RA of chitinase; Y, X, and $T_{out}$), which makes it difficult to choose the best condition. An alternative was to use the models developed in the CCRD (Table 5) to simultaneously optimize the process responses using the global desirability function proposed by Derringer and Suich [55]. Table 6 presents the values selected for the levels of each response, as well as the desirability associated with each level. The desirability profiles for the spray drying process are shown in Figure 4.

**Table 6.** Desirability values selected for the dependent variables (responses) of the spray drying of enzyme extract.

| Dependent Variables | Level | Value | Desirability * |
|---|---|---|---|
| $RA_{β-1,3-glucanase}$ (%) | Low (L) | 26.71 | 0.00 |
| | Medium | 64.38 | 0.50 |
| | High (H) | 102.05 | 1.00 |
| $RA_{chitinase}$ (%) | Low (L) | 0.00 | 0.00 |
| | Medium | 27.93 | 0.50 |
| | High (H) | 55.86 | 1.00 |
| Y (%) | Low (L) | 26.90 | 0.00 |
| | Medium | 36.79 | 0.50 |
| | High (H) | 46.68 | 1.00 |
| $X_p$ (%) | Low (L) | 1.80 | 1.00 |
| | Medium | 4.20 | 0.50 |
| | High (H) | 6.60 | 0.00 |
| $T_{out}$ (°C) | Low (L) | 62.00 | 1.00 |
| | Medium | 76.50 | 0.50 |
| | High (H) | 91.00 | 0.00 |

$RA_{β-1,3-Glucanase}$: β-1,3-Glucanase residual enzymatic activity; $RA_{chitinase}$: chitinase residual enzymatic activity; Y: powder recovery yield; $X_p$: powder moisture content; $T_{out}$: outlet air temperature. * The value 0.00 refers to an undesirable response, while the value 1.00 refers to a highly desirable response.

According to the analysis of the desirability profiles, the optimum spray drying conditions were Tin of 120 °C, Q of 1.1 $m^3$/min, F of 5.8 mL/min and P of 0.4 MPa. These values correspond to the CCRD central point. In the assays using the optimized spray drying conditions, a final product with an RA of 94 and 54% for β-1,3-glucanase and chitinase, respectively, was obtained. In addition, the yield was 39%, with an $X_p$ of 4% and a $T_{out}$ of 74 °C. As the global desirability value was 0.69, all responses were within the desirability limits.

Spray drying in optimized conditions showed good results of RA of β-1,3-glucanase and $X_p$. The values obtained for RA of chitinase were promising, but there is a need to evaluate the use of other ranges of operational parameters for this enzyme. The yield obtained in the optimum conditions was in agreement with those of laboratory scale spray dryers.

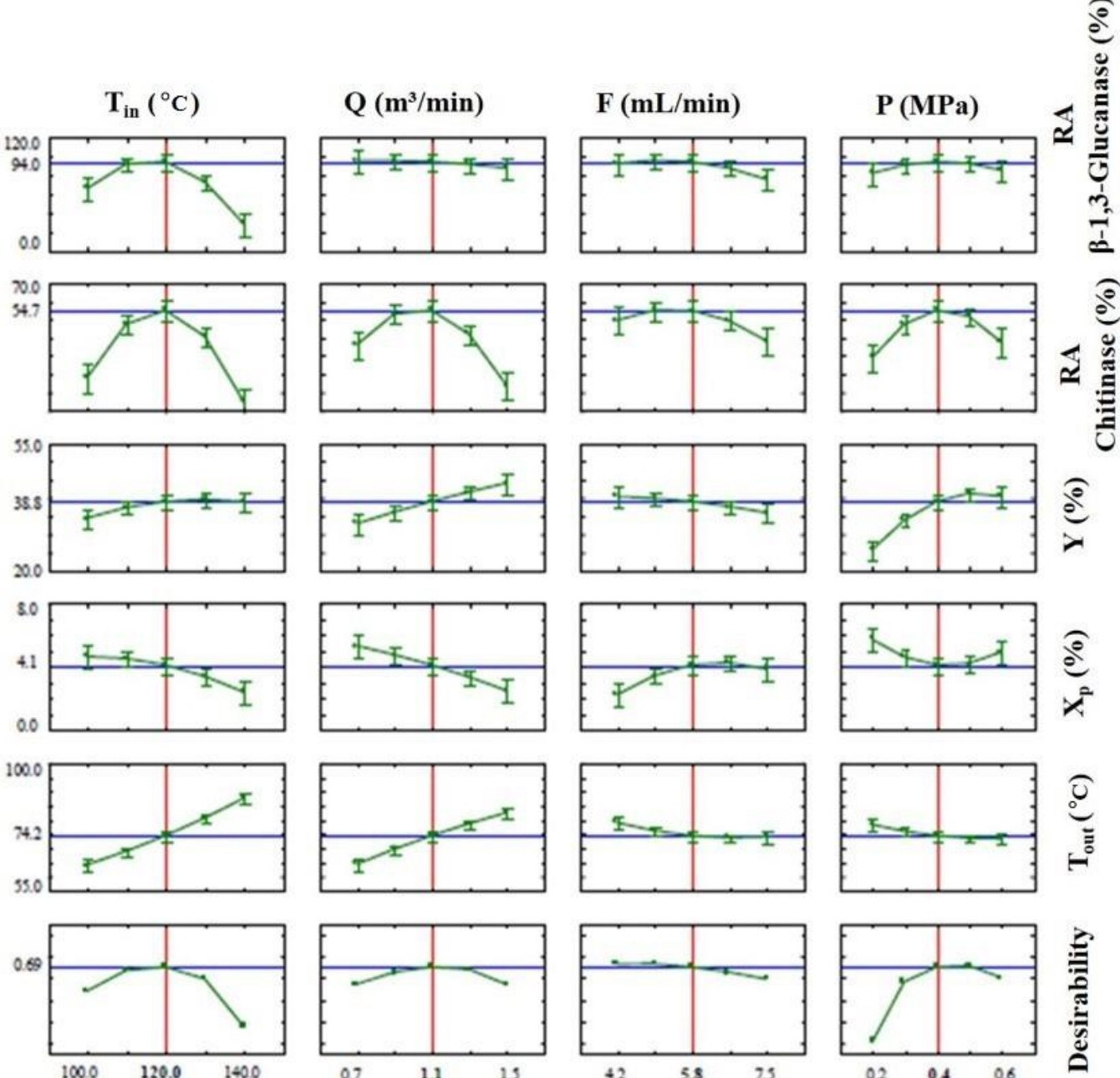

**Figure 4.** Desirability profiles of the spray drying optimization. $T_{in}$: inlet air temperature; Q: drying air flow rate; F: feed flow rate; P: nozzle air pressure; $RA_{\beta\text{-}1,3\text{-Glucanase}}$: β-1,3-Glucanase residual enzymatic activity; $RA_{chitinase}$: chitinase residual enzymatic activity; Y: powder recovery yield; $X_P$: Powder moisture content; $T_{out}$: outlet air temperature.

### 3.4. Insecticidal Activity of the Spray-Dried Powder

Figure 5 shows that the maximum control efficiency 24 h after the emergence of *C. capitata* pupae was 65.6% and it was obtained in the treatment with a powder concentration of 100 g/L. Similar results were found in other studies on the biocontrol of *C. capitata*. Lozano-Tovar et al. [56], using the crude fermented extract of *Metarhizium brunneum* and *Metarhizium roberstii*, obtained control efficiencies (24 h after the emergence of the fly) of approximately 70 and 60%, respectively. Yousef et al. [57] reported a control efficiency of 73% (25 h after the emergence of the fly) using the crude extract of *M. brunneum*. However, the runs of insecticidal activity were performed by ingestion and not by contact, as the

current work did. The worst control efficiency (6.3%) was obtained with the treatment of the highest powder concentration (200 g/L). The solution prepared with this concentration showed a viscous aspect with high turbidity. This probably repelled the insects and prevented their contamination, resulting in poor control efficiency.

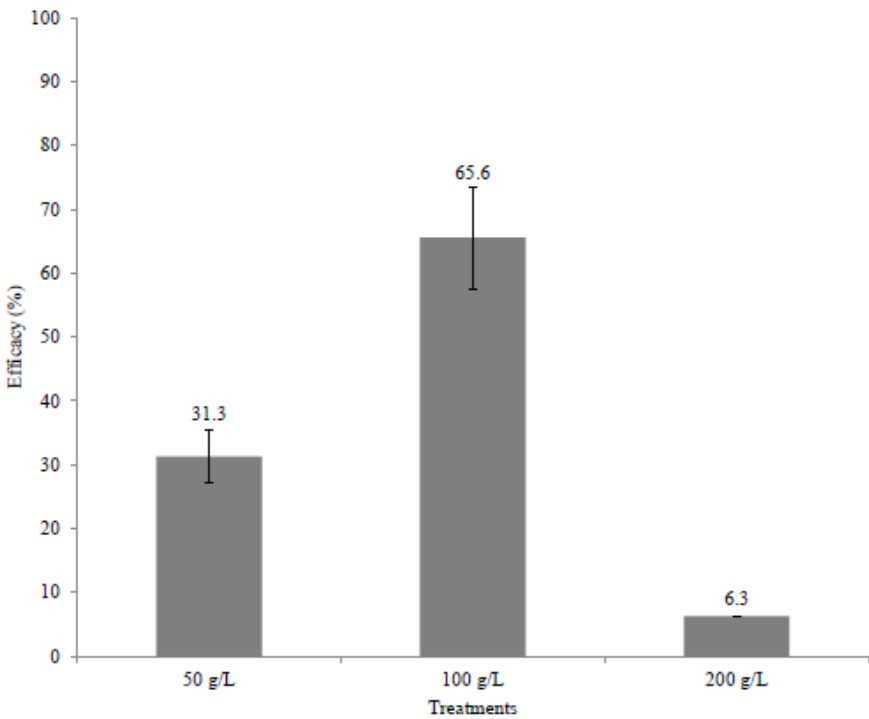

**Figure 5.** Efficacy of the spray-dried powder containing hydrolytic enzymes in the control of *Ceratitis capitata* 24 h after emergence.

## 4. Conclusions

Gum arabic and maltodextrin DE20 were the carriers used in the spray drying of enzymatic extract with the highest residual enzymatic activity of chitinase and β-1,3-glucanase. Soluble starch was the carrier with the highest powder recovery yield. The carrier combination of maltodextrin (2.5% *w/v*), gum arabic (2.5% *w/v*), and soluble starch (5.0% *w/v*) showed the best results of residual enzymatic activity of β-1,3-glucanase (88.36%) and chitinase (69.82%), and a powder recovery yield of 45.49%. The best spray drying conditions were obtained using an air inlet temperature of 120 °C, drying airflow rate of 1.1 m³/min, feed flow rate of 5.8 mL/min, and nozzle air pressure of 0.4 MPa. The powder produced in these conditions showed 65.6% efficiency in controlling the fly *C. capitata*. These results demonstrated the possibility of using spray drying to obtain a final product with a high chitinase and β-1,3-glucanase activity and its potential use in biological control. This effectiveness obtained in the laboratory indicates two paths to follow. The first is to further study the promising potential of the bioproduct obtained for the control of this pest, and the second is the need to take these tests to orchards in the field.

**Author Contributions:** Conceptualization, B.C.A. and M.A.M.; Data curation, M.V.T.; Formal analysis, T.B., S.S., A.F. and V.F.B.; Methodology, T.B., S.S., L.d.A.C., G.C., D.E.N. and J.V.C.G.; Supervision, M.A.M.; Writing—original draft, G.L.Z., R.C.K. and M.A.M.; Writing—review and editing, M.V.T., R.C.K. and M.A.M. All authors have read and agreed to the published version of the manuscript.

**Funding:** This study was supported by the Research Support Foundation of the State of Rio Grande do Sul (FAPERGS: 17/2551-0000893-6; 21/2551-0002253-1), Coordination for the Improvement of Higher Education Personnel (CAPES: 001), National Council for Scientific and Technological Development (CNPq: 428180/2018-3; 306241/2020-0), and Catarinense Federal Institute of Science, Technology and Education (IFC: 128/2018).

**Institutional Review Board Statement:** Not applicable.

**Informed Consent Statement:** The strain of *Ceratitis capitata* used in the experiments comes from the Brazilian Agricultural Research Corporation (EMBRAPA) collection, licensed at National System for the Management of Genetic Heritage and Associated Traditional Knowledge (SISGEN) for access to the Genetic Heritage (AA0E5E8) for this purpose. In the case of insects, authorization from the ethics committee does not apply.

**Conflicts of Interest:** The authors declare no conflict of interest.

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
