# Peer review of "Spray-Dried Powder Containing Chitinase and β-1,3-Glucanase with Insecticidal Activity against Ceratitis capitata (Diptera: Tephritidae)"

_processes, doi:10.3390/pr10030587_

Round 1

Reviewer 1 Report

The manuscript entitled "Spray-dried powder containing chitinase and β-1,3-glucanase with insecticidal activity against Ceratitis capitata (Diptera: Tephritidae)" describes the use of spray drying method to obtain chitinase and β-1,3-15 glucanase as active ingredients for the biological control of agricultural pests. The paper is well written, the methodology is correct and the results are proven by experiments. Based on the above, I  recommend the publication of the manuscript with a minor correction: the scale bar length in SEM images should be more visible. 

Author Response

The manuscript entitled "Spray-dried powder containing chitinase and β-1,3-glucanase with insecticidal activity against Ceratitis capitata (Diptera: Tephritidae)" describes the use of spray drying method to obtain chitinase and β-1,3-15 glucanase as active ingredients for the biological control of agricultural pests. The paper is well written, the methodology is correct and the results are proven by experiments. Based on the above, I  recommend the publication of the manuscript with a minor correction: the scale bar length in SEM images should be more visible.

Answer: We would like to thank the Reviewer 1 for their interest in our work and for helpful comments that greatly improves the manuscript. As indicated above, we have altered the sale bar length in SEM images.

Reviewer 2 Report

I affirm that the quality and significance of the results presented, the manuscript standard of writing, and the overall presentation of the content is up to scientific standard. Nevertheless, I could not approve the paper for acceptance in its current form as I have found some minor errors that are needed to be corrected before the paper could be reconsidered for acceptance. For instance, several of the scientific names were not italicized; see L55, 61-62, 70, 154, 157, 166, 194, 450, 453, 454, 457, 464, and 475. The authors are encouraged to carefully look through the entire manuscript to ensure this error is corrected before the manuscript is resubmitted.

In addition, there is also a need to amend some grammatical errors, incomplete statements, and inappropriate expressions, here and there in the text. I have highlighted some in the specific comments section.

I noticed the manuscript is missing the discussion section. I understand in this journal (Processes), it is not mandatory to include this part, however, I assume the authors would have been able to beautifully link/contrast their results and the significance of the current study to some of the previously conducted related studies. If the authors can, just my opinion though, not obligatory, I like to suggest that they could make section 3; Results and discussion, then, they would be able to add 1-2 statements about the relevance of their findings in respect of what has been reported previously in the literature.

Please, find the attached file for more specific comments.

Author Response

ANSWER TO REVIEWER 2

  1. I affirm that the quality and significance of the results presented, the manuscript standard of writing, and the overall presentation of the content is up to scientific standard. Nevertheless, I could not approve the paper for acceptance in its current form as I have found some minor errors that are needed to be corrected before the paper could be reconsidered for acceptance. For instance, several of the scientific names were not italicized; see L55, 61-62, 70, 154, 157, 166, 194, 450, 453, 454, 457, 464, and 475. The authors are encouraged to carefully look through the entire manuscript to ensure this error is corrected before the manuscript is resubmitted. In addition, there is also a need to amend some grammatical errors, incomplete statements, and inappropriate expressions, here and there in the text. I have highlighted some in the specific comments section.

Answer: We thank the Reviewer 2 for their interest in our work and for helpful comments that greatly improves the manuscript. We have checked all the general and specific comments provided by the Reviewer and have made the necessary changes accordingly to their indications.

  1. I noticed the manuscript is missing the discussion section. I understand in this journal (Processes), it is not mandatory to include this part, however, I assume the authors would have been able to beautifully link/contrast their results and the significance of the current study to some of the previously conducted related studies. If the authors can, just my opinion though, not obligatory, I like to suggest that they could make section 3; Results and discussion, then, they would be able to add 1-2 statements about the relevance of their findings in respect of what has been reported previously in the literature.

Answer: We understand the suggestion made by the Reviewer, but a separated section of discussion is difficult specifically for this manuscript. In this sense, we have highlighted and/or added new discussion of the results through the section Results and discussion.

  1. Please, find the attached file for more specific comments.

Answer: All suggestions made by the Reviewer were accepted. See the highlighted modifications in the revised version of the manuscript.